# Temporal variability predicts the magnitude of between-group attentional blink differences in developmental dyslexia: a meta-analysis

Nicholas A. Badcock and Joanna C. Kidd

ARC Centre of Excellence in Cognition and its Disorders, Department of Cognitive Science, Macquarie University, North Ryde, New South Wales, Australia

## ABSTRACT

**Background.** Here we report on a meta-analysis of the between-group main effect (Group Difference) noted in the attentional blink (AB) research focused on specific reading impairment, commonly referred to as developmental dyslexia. The AB effect relates to a limitation in the allocation of attention over time and is examined in a dual-target rapid serial visual presentation (RSVP) paradigm. When the second target appears in close temporal proximity to the first target, the second target is reported less accurately.

**Method.** A Web of Science search with terms "attentional blink" & dyslexia returned 13 AB experiments (11 papers) conducted with developmental dyslexia. After exclusions, 12 experiments were included in the meta-analysis. The main pattern of performance from those experiments was lower overall accuracy in groups of individuals with dyslexia relative to typically reading peers; that is, a between-group main effect. This meta-analysis examined the size of the Group Difference in relation to temporal and task-set related features, which differed between and within experiments.

**Results.** Random effects modelling indicated a significant Group Difference of $-0.74$ standard deviation units, 95% CI $[-.96, -.52]$, $p < .001$ (excluding one anomalous result): implicating significantly poorer overall dual-target performance in dyslexic readers. Meta-regression analyses indicated two variables related to the Group Difference; pre-RSVP time and temporal variability of the second target relative to the first target within the RSVP.

**Discussion.** It is suggested that the endogenous engagement of the temporal features of task-set is slower or disrupted in developmental dyslexia.

Corresponding author
Nicholas A. Badcock,
nicholas.badcock@mq.edu.au

## INTRODUCTION

Aside from a specific difficulty with the typical acquisition of reading, developmental dyslexia has been associated with a number of cognitive weaknesses. One of these weaknesses is the ability to rapidly deploy attention across time (*Tallal, 1976*; e.g., *Hari & Renvall, 2001*). Here we focus on a single paradigm used to assess the allocation of visual

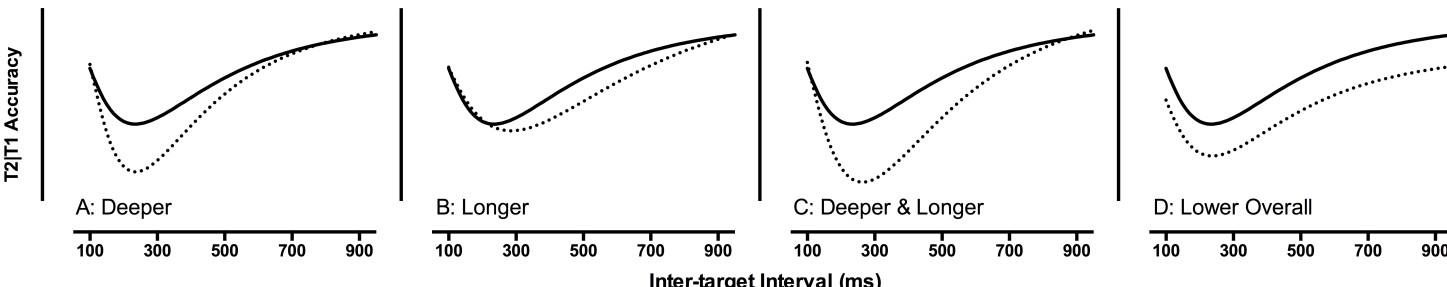

**Figure 1 Four patterns of attentional blink (AB) performance.** Second target performance, given that the first target was correctly reported, as a function of the inter-target interval. The solid line depicts a standard AB performance. The dashed lines depict atypical AB performance related to deeper (A), longer (B), and deeper and longer AB effects (C), as well as lower overall accuracy (D).

attention across time: a dual-target Rapid Serial Visual Presentation (RSVP) paradigm. Performance in this paradigm has been described as an 'Attentional Blink' (AB). The AB is an attentional phenomenon whereby the processing of the first target (T1) is considered to disrupt the processing of a second target (T2) when the two targets appear in close temporal proximity (i.e., within 500 ms; *Broadbent & Broadbent, 1987*; *Raymond, Shapiro & Arnell, 1992*). The standard AB pattern is illustrated by the solid line in Fig. 1. The main point to note is that, at short inter-target intervals (e.g., 200–300 ms), T2 accuracy is lower than at long (e.g., 500–700 ms) intervals.

In a review of the AB literature on developmental dyslexia, *McLean et al. (2010)* demonstrated that the most common difference between dyslexic and typically reading groups was a main effect; that is, overall, the performance of the dyslexic readers was lower than for typical readers. Therefore, rather than a difficulty in rapidly deploying attention across time (or "sluggish attentional shifting", see *Hari & Renvall, 2001*), dyslexic readers had a general difficulty with the AB paradigm. To illustrate this point, four different patterns of AB performance are presented in Fig. 1. Specific difficulties with the AB may relate to deeper, longer, or deeper and longer effects (illustrated in Fig. 1A–1C). However, what is noted in the dyslexia literature is the fourth option, lower overall accuracy (see Fig. 1D), reflecting a general difficulty with the dual-target paradigm (note: unless otherwise specified, references to 'single-target' and 'dual-target' refer to RSVP procedures). The current paper reports on a meta-analysis of the AB and dyslexia literature to explore this general difficulty.

The AB is considered to reflect a temporal limitation of attention. Most generally it is thought that a finite set of resources is required to consolidate a short-term representation of a target item for conscious report. While these resources are consolidating of T1, the representation of T2 may decay and will not be available for conscious report (for reviews see *Dux & Marois, 2009*; *Martens & Wyble, 2010*). If this consolidation lasts longer, short-term representations waiting to be processed may decay; in order to update these representations for consolidation, regressive eye-movements made during reading may increase, although regressions may have other explanations as well (e.g., checking for comprehension). This pattern of eye-movements has been noted in developmental dyslexia (*Schneps et al., 2013*). However, of the thirteen experiments published in this area,

only three identify AB performance consistent with longer consolidation (or more rapid memory decay). These three experiments (*Hari, Valta & Uutela, 1999*; *Lallier et al., 2010*; *Laasonen et al., 2012*) indicate that the accuracy of T2 report in dyslexia reaches the same level of accuracy as that of typical readers at long inter-target intervals; however, the time required to reach this accuracy is longer for dyslexic than typical readers. The majority of the experiments do not demonstrate this interaction, with accuracy remaining lower in dyslexic readers across all inter-target intervals (*Visser, Boden & Giaschi, 2004*; *Buchholz & Aimola Davies, 2007*; *Badcock, Hogben & Fletcher, 2008*; *Facoetti et al., 2008*; *Lallier, Donnadieu & Valdois, 2010*; *McLean et al., 2010*). One anomalous experiment reports higher accuracy across all inter-target intervals in a group of dyslexic readers (*Lacroix et al., 2005*); this will be considered in the discussion. If the majority of the evidence points to a general deficit in the AB paradigm for dyslexic readers, what underpins the general deficit?

When *McLean et al. (2010)* added single-target accuracy as a covariate in their analysis, twenty per cent of the dual-target performance difference between dyslexic and typical readers was accounted for. When controlling for single-target accuracy, in combination with a continuous-performance measure (accounting for nine per cent of the between-group variation), the between-group effect was no longer significant. Simulating this sort of inattention factor produces patterns of data that mimic dyslexic group performance (*Stuart, McAnally & Castles, 2001*; *Roach, Edwards & Hogben, 2004*). A general factor common to single- and dual-target paradigms may account for the between group differences noted in the AB and also have much broader implications beyond RSVP paradigms.

In RSVP paradigms, the target serial position varies in time relative to the onset of the RSVP. Single-target performance has been shown to be sensitive to this temporal variation. In a series of experiments, *Ariga & Yokosawa (2008)* demonstrated that single-target accuracy increased as a function of foreperiod; that is, at longer intervals from the onset of the RSVP, accuracy was higher. There is a body of literature on the effects of temporal orienting, particularly with respect to reaction time (*Niemi & Näätänen, 1981*; *Nobre, Correa & Coull, 2007*), but temporal predictability and cueing have also been demonstrated to increase accuracy in the AB (cueing *Martens & Johnson, 2005*; predictability *Badcock et al., 2013*). Recently, *Tang, Badcock & Visser (2014)* demonstrated that dual-target practice improvements are attributable to learning the target temporal locations. Given that temporal orienting has a role in both single- and dual-targets, it presents as a candidate explanation for differences between dyslexic and typical readers in RSVP performance.

Another element common to single- and dual-target paradigms is the engagement of 'task-set.' Monsell proposed the concept of task-set with respect to task-switching paradigms (*Rogers & Monsell, 1995*; *Monsell, 1996*) whereby, even during a simple paradigm when the task was well known to the participants, a cognitive model of the task-requirements must be engaged to complete the task. In an RSVP paradigm, the cognitive model includes searching for letters while ignoring numbers and reporting the identity of letters. Therefore, this concept is similar to the proposal that endogenous control of a 'visual-filter' is set up at the outset of the RSVP task (*Di Lollo et al., 2005*). Di Lollo et al. suggest that the AB is caused by a 'temporal loss of control' of the visual filter

(but see *Dell'Acqua et al., 2009*; and also *Olivers et al., 2011*). The engagement of task-set may account for time-related increases in single-target accuracy (i.e., *Ariga & Yokosawa, 2008*). Nevertheless, it is an additional candidate for the differences between dyslexic and typical readers noted in RSVP performance.

The current investigation aimed to explore the basis of the significantly lower dual-target accuracy that has been reported in dyslexic versus typical reading groups. Based on evidence that single-target performance can account for the dual-target between-group difference, we examined variability in temporal and task-set related features via a meta-analysis of the AB literature on developmental dyslexia. Temporal variations and task-set requirements were selected as common to both single- and dual-target tasks.

## METHOD

### Experiment selection

Searching the Web of Science for "attentional blink" & dyslexia returned 26 entries (23rd of September 2014). Papers were required to include a comparison of dyslexic readers with respect to age-matched typical readers on a dual-target task requiring the identification and/or detection of two targets: 12 papers were relevant, two of which included two experiments (*Visser, Boden & Giaschi, 2004*; *Buchholz & Aimola Davies, 2007*). *Badcock, Hogben & Fletcher (2011)* report a reanalysis of their 2008 data. This was not included in the present meta-analysis because it is not an independent experiment. *McLean et al. (2011)* report an extension of their (2010) research, doubling the sample size. The 2010 work was therefore excluded from the analyses. The comprehensiveness of this set of experiments was confirmed using PubMed and PsycInfo using the same search terms, which returned the same set of experiments.

The final number of experiments included was 12 (*Hari, Valta & Uutela, 1999*; Experiments 1 and 2, *Visser, Boden & Giaschi, 2004*; *Lacroix et al., 2005*; Experiment 1 and 2, *Buchholz & Aimola Davies, 2007*; *Badcock, Hogben & Fletcher, 2008*; *Facoetti et al., 2008*; *Lallier et al., 2010*; *Lallier, Donnadieu & Valdois, 2010*; *McLean et al., 2011*; *Laasonen et al., 2012*).

As in any field, a bias for the publication of significant results may mean that this meta-analysis overestimates the true between-group difference (i.e., "the file drawer problem" *Rosenthal, 1979*). However, as the AB and dyslexia research has focused on a between-group interaction, bias in not reporting the between-group main effect should be reduced. The main objective of this analysis is to examine variables related to this between-group difference. Noteworthy relationships will need to be directly manipulated in dyslexia investigations prior to theoretical incorporation. This meta-analysis takes the additional step of exploring the meta-analytic variable with respect to task-parameters. The PRISMA checklist (*Moher et al., 2009*) is included in Supplemental Information 1.

### Variable selection and calculation

In addition to the Group difference, seven variables were included as possible moderators: pre-RSVP time; stimulus onset asynchrony; T1, T2, and RSVP temporal variation, total

**Table 1 Parameter descriptive statistics, normality tests, and values for each experiment on the attentional blink and dyslexia.** Descriptive statistics are based on $n = 11$ experiments, excluding *Lacroix et al. (2005)* and *McLean et al. (2010)*, except for the Pre-RSVP time variable that is based on $n = 8$ experiments, excluding experimental tasks initiated with a key press.

| Statistic | | | Pre-RSVP time[a] | SOA | T1 time difference | T2 time difference | RSVP time difference | Total identities | Task complexity | Group difference |
|---|---|---|---|---|---|---|---|---|---|---|
| Mean (SD) | | | 600 (316) | 105.67 (10) | 621.99 (353) | 930.92 (315) | 1179.07 (854) | 13.82 (7) | 1.64 (1) | −0.98 (0) |
| Median (IQR) | | | 600 (200) | 100 (6.55) | 424 (593.25) | 900 (615.95) | 958.5 (737.5) | 12 (9.5) | 1 (2.5) | −0.89 (0.56) |
| Min | | | 0 | 100 | 125 | 600 | 0 | 6 | 0 | −1.97 |
| Max | | | 1,000 | 133 | 1,066 | 1,333 | 3,066 | 25 | 3 | −0.39 |
| Shapiro–Wilk | | | 0.87 | 0.63[**] | 0.85[*] | 0.83[*] | 0.9 | 0.88 | 0.77[**] | 0.9 |
| **Authors** | **Year** | **Exp.** | | | | | | | | |
| Hari et al. | 1999 | 1 | 0 | 106.5 | 958.5 | 1171.5 | 958.5 | 25 | 3 | −0.39 |
| Visser et al. | 2004 | 1 | key press | 100 | 300 | 1,300 | 1,600 | 6 | 0 | −0.77 |
| Visser et al. | 2004 | 2 | key press | 100 | 300 | 1,300 | 1,600 | 6 | 1 | −0.89 |
| *Lacroix et al.* | *2005* | *1* | *1,000* | *100* | *400* | *700* | *0* | *10* | *1* | *1.12* |
| Buchholz & Aimola Davies | 2007 | 1 | 1,000 | 100 | 400 | 600 | 0 | 9 | 1 | −1.97 |
| Buchholz & Aimola Davies | 2007 | 2 | 1,000 | 100 | 400 | 600 | 0 | 17 | 0 | −0.94 |
| Badcock et al. | 2008 | 1 | 500 | 100 | 900 | 1,100 | 1,600 | 20 | 3 | −0.92 |
| Facoetti et al. | 2008 | 1 | 600 | 100 | 125 | 900 | 925 | 9 | 0 | −0.66 |
| Lallier et al. | 2010 | 1 | 600 | 116 | 928 | 696 | 928 | 12 | 3 | −1.47 |
| Lallier et al. | 2010 | 2 | 600 | 100 | 1,040 | 600 | 800 | 12 | 3 | −1.62 |
| *McLean et al.* | *2010* | *1* | *500* | *106.6* | *424* | *639.6* | *1492.4* | *11* | *1* | *−0.79* |
| McLean et al. | 2011 | 1 | 500 | 106.6 | 424 | 639.6 | 1492.4 | 11 | 1 | −0.62 |
| Laasonen et al. | 2012 | 1 | key press | 133.3 | 1066.4 | 1,333 | 3065.9 | 25 | 3 | −0.51 |

**Notes.**

SOA, Stimulus Onset Asynchrony; T1/T2, first/second target.

T2 time difference is relative to T1 temporal position, Group Difference is Cohen's d in standard deviation units.

[*] $p < .05$.

[**] $p < .01$.

[a] $n = 8$.

target and distractor identities, and task complexity. All data points for each study, including *McLean et al. (2010)*, are included in Table 1.

### Group difference

The Group Difference was estimated as the effect size of between-group difference (dyslexic minus typical) for overall mean T2 accuracy, given that T1 was correctly reported (T2| T1). This is the average accuracy at each of the inter-target intervals. The associated variance, required for the meta-analysis, was the standard deviation of this accuracy at each inter-target interval. The Group Difference was calculated in standard deviation units, Cohen's d, using an unbiased estimation (*Cumming, 2012*; *Cumming, 2014*) with the 'escalc' function in 'metafor' (*Viechtbauer, 2010*). In all but two cases, the mean and standard deviation for the dyslexic and control reading groups were supplied by authors, if it was not available in the published manuscripts (available in *Lallier et al., 2010*). Where the required data were not reported or could not be obtained

(*Hari, Valta & Uutela, 1999*; *Lacroix et al., 2005*), the data were estimated from the published figures using WebPlotDigitizer (*Rohatgi, 2014*).

### Pre-RSVP time

Pre-RSVP time is defined as the time period (in ms) before the presentation of the first RSVP item, typically constituting the inter-trial interval. This includes the duration of a fixation symbol and any fixation to RSVP time where no stimulus was presented. In three experiments (*Visser, Boden & Giaschi, 2004*; *Laasonen et al., 2012*), the fixation symbol remained on screen until a key-press. These experiments were excluded on the basis that the presentation duration of the fixation could not be determined. *Hari, Valta & Uutela*'s (*1999*) experiment presented no fixation symbol, therefore forming a baseline condition where the first element of the presentation is a distracter item. The final number of experiments included for this variable was eight, after the *Lacroix et al. (2005)* experiment was excluded based on the test of homogeneity (see 'Results').

### Stimulus onset asynchrony (SOA)

SOA represents the time period between the onset of one stimulus and the next. This was determined from the method sections of the respective papers.

### T1, T2, and RSVP temporal variation

Temporal variation was calculated as the difference between the minimum and maximum temporal presentation values of T1 within the RSVP and T2 relative to T1, and the longest minus the shortest RSVP sequences.

For example, if T1 is presented at RSVP positions 6, 7, and 8, and the stimulus onset asynchrony is 100 ms, this would correspond to 600 (minimum), 700, and 800 (maximum) ms, with a difference of 200 ms. If T2 is presented at positions 1–12 following T1, this would correspond to 100 through to 1,200, with a difference of 1,100 ms. If two distractor items always follow T2 in the RSVP, then the minimum RSVP would be 900 (min T1 + min T2 + 2 = 600 + 100 + 200), the maximum 2,200 (max T1 + max T2 + 200 = 800 + 1,200 + 200), with a difference of 1,300 ms.

In the case of *Facoetti et al. (2008)* where only two targets and accompanying masks were presented, target and RSVP timings corresponds to the number of temporal positions starting with the earliest time of T1 presentation following fixation (125 ms). It is worth noting that this 'skeletal' RSVP paradigm may produce more variable results at the electrophysiological level (see *Craston, Wyble & Bowman, 2006*).

### Total identities

By 'identity' we refer to the number of standard verbal labels of T1, T2, and distractors. For example, if T2 is a fixed letter of the alphabet (e.g., letter X), and T1 as well as the distractors are any letter other than that used for T2, there is 1 identity for T2, 25 possible identities for T1, and 25 for the distractors: with the total number of identities being 26. Note: T1 and the distractor identities both have 25 possibilities because they are randomly selected on each trial and the T1 identity for one trial will be the distractor identity for another trial. This example is for illustrative purposes, many studies omit 'I,' 'O,' and 'Q' due to the limited masking properties.

The *Visser, Boden & Giaschi (2004)* and *McLean et al. (2011)* experiments included random-dot distractors with different 'identities' for each presentation. Although the precise number of identities is difficult to determine, this was classified as a single identity given that independent semantic labels could not be applied to these variations.

### Task complexity

Task complexity was calculated as the sum of following criteria. (1) A switch between T1 and T2. This included a change in decision type (identification to detection), a change in stimulus search (e.g., any white letter to a black letter X), a change in stimulus category (e.g., arrows to shapes), or a change in spatial location. (2) Whether the target and distractor items were from the same stimulus category. One point was assigned for each of the five criteria that were met.

### Analysis

The between-group effect-size (i.e., Group Difference) across studies was estimated using a random effect model implemented in R using the 'metafor' package (*Viechtbauer, 2010*). The relationships between the dependent measures were tested using point estimate and rank correlations (Pearson's product-moment and Spearman's *rho*) and moderation was tested using random effects modelling. Both parametric and non-parametric correlations are reported to be conservative with respect to the small number of studies.

### Not correcting for multiple comparisons

Correlations were calculated to examine the relationships between the dependent measures in this paper. We did not correct for multiple comparisons in this instance. This decision was made to reduce the risk of Type 2 errors and on the basis that relationships of note in the meta-analysis will need to be demonstrated empirically before they are applied theoretically (see *Cabin & Mitchell, 2000*, for a discussion on considering correcting for multiple comparisons).

## RESULTS

The Group Difference for each study is presented in a forest plot in Fig. 2. The overall difference was $-0.70$, $p = 0.001$, indicating that across the 12 experiments, overall T2|T1 accuracy was poorer in dyslexic readers. A test of heterogeneity indicated that 57% of the variation was due to between study differences, $Q_{(11)} = 23.6$, $p = .015$. This was entirely attributable to the *Lacroix et al. (2005)* experiment, its exclusion leading to a non-significant test for heterogeneity: 0%, $Q_{(10)} = 9.8$, $p = .46$. Thus the Lacroix et al. result is sufficiently different that it may be considered sampled from a separate population. For this reason, it was excluded from subsequent analyses. The overall estimate of effect size excluding this experiment was $-0.74$, 95% CI $[-.96, -.52]$, $p < .001$.

Descriptive statistics for the 8 variables are presented in Table 1 (data points for all experiments are also presented). The SOA, T1 and T2 time difference, and total identities variables were not normally distributed. Therefore, Spearman non-parametric correlations are more appropriate indices of the relationships for these variables, and have been reported alongside Pearson product-moment values in Table 2. Most critical is the bottom

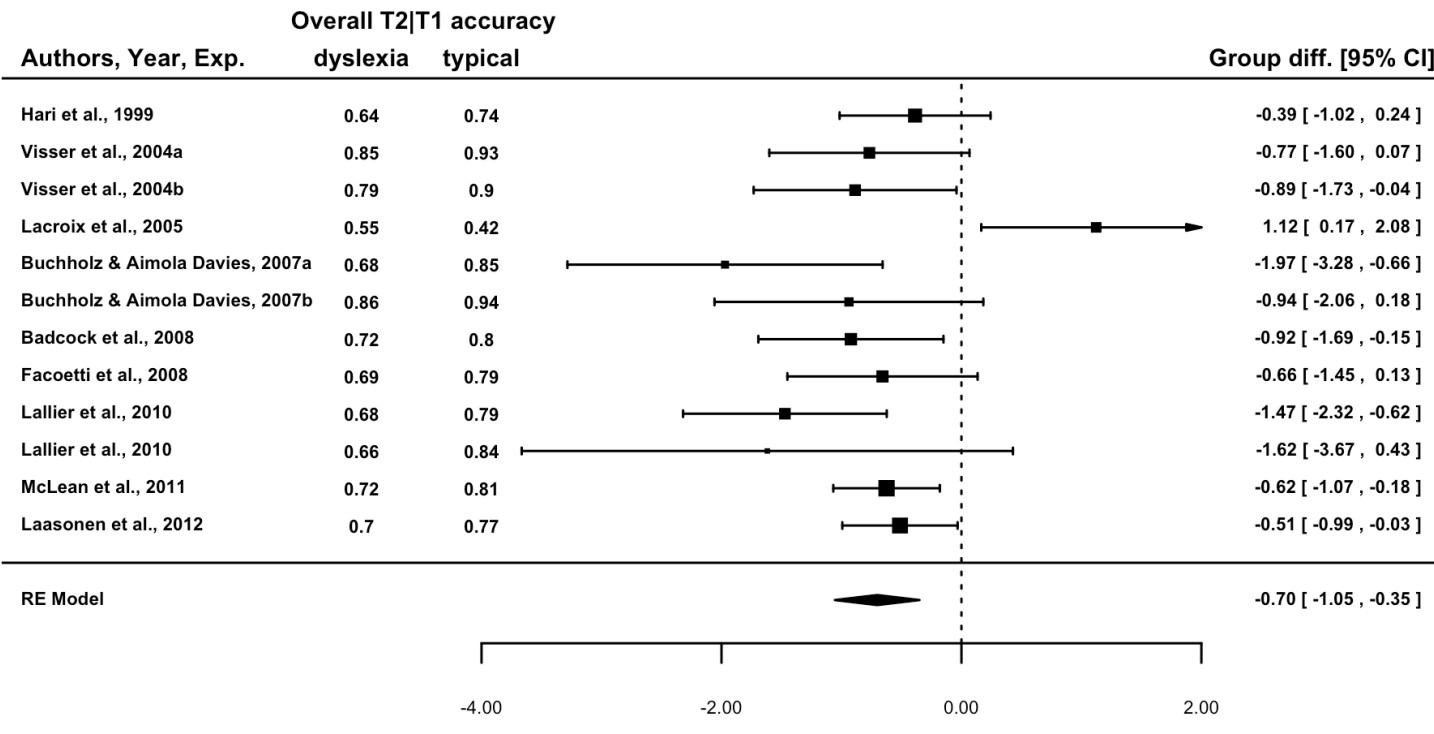

**Figure 2  Forest plot of the group difference in overall T2|T1 accuracy by experiment.** Data point size is based upon the inverse of the variance (i.e., larger points correspond to smaller variance). T2|T1 denotes T2 accuracy calculated for those trials in which T1 was correctly reported.

**Table 2  Correlations coefficients (Pearson: Spearman) between variables of the attentional blink experiments on developmental dyslexia ($n = 11$).**

| Parameter | 1 | 2[a] | 3[a] | 4[a] | 5 | 6 | 7[a] |
|---|---|---|---|---|---|---|---|
| 1. Pre-RSVP time[b] | | | | | | | |
| 2. SOA | $-0.32 : -0.33$ | | | | | | |
| 3. T1 time difference | $-0.43 : -0.26$ | $0.56 : 0.62^*$ | | | | | |
| 4. T2 time difference | $-0.77^* : -0.87^{**}$ | $0.28 : 0.29$ | $0.07 : 0.01$ | | | | |
| 5. RSVP time difference | $-0.68 : -0.82^*$ | $0.66^* : 0.26$ | $0.30 : 0.01$ | $0.73^* : 0.88^{**}$ | | | |
| 6. Total identities | $-0.56 : -0.34$ | $0.55 : 0.50$ | $0.71^* : 0.80^{**}$ | $0.23 : 0.04$ | $0.30 : -0.01$ | | |
| 7. Task complexity | $-0.53 : -0.32$ | $0.50 : 0.52$ | $0.96^{**} : 0.88^{**}$ | $0.11 : 0.13$ | $0.32 : 0.12$ | $0.63^* : 0.66^*$ | |
| 8. Group difference | $-0.66 : -0.86^{**}$ | $0.26 : 0.47$ | $-0.06 : 0.08$ | $0.62^* : 0.68^*$ | $0.58 : 0.62^*$ | $0.41 : 0.28$ | $-0.09 : 0.01$ |

**Notes.**

SOA, Stimulus Onset Asynchrony; T1/T2, first/second target; rel T1, relative to T1.

[a] non-normally distributed variable

[b] $n = 8$.

[*] $p < .05$.

[**] $p < .01$.

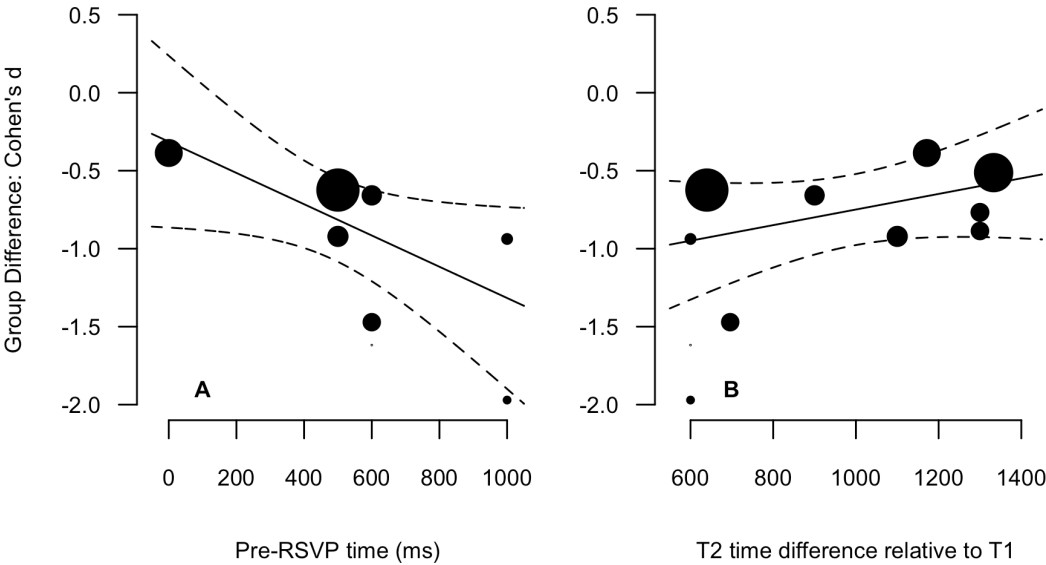

**Figure 3 Scatter plots and linear regression fits for the Group Difference (Cohen's d, *y*-axis) again pre-RSVP time (ms, A) and T2 time difference relative to T1 (ms, B).** Data point size is based upon the inverse of the variance (i.e., larger points correspond to smaller variance), solid lines are regression fits, and dashed lines are 95% confidence intervals.

row in which correlations for the relationship of Group Difference to all other variables are reported. There are three relationships worth drawing attention to. The first is a strong negative relationship between Pre-RSVP time and the Group Difference (Pearson $= -.66$, Spearman $= -.86$), indicating that the greater the time before the onset of the RSVP, the greater the between-group accuracy discrepancy. The second two are the strong positive correlations between T2 time difference relative to T1 (Pearson $= .62$, Spearman $= .68$) and the RSVP time difference (Pearson $= .58$, Spearman $= .62$) and the Group Difference. These relationships indicate that the greater the temporal variability of T2 relative to T1 and within the overall RSVP, the smaller the between-group difference. It is important to note that all three of these variables are highly correlated with absolute magnitudes between .68 and .88. The relationships between the Group Difference and Pre-RSVP time and T2 time difference relative to T1 are presented in scatter plots with linear fits in Fig. 3. The relationship between the Group Difference and Pre-RSVP time was the only significant moderating relationship: $\beta = -0.001$ $[-0.002, 0.000]$, $Q_{(1)} = 3.87$, $p = 0.049$. This indicates that a 1 msec increase in pre-RSVP time increases the group difference by 0.001 standard deviations units; an additional 500 msec before RSVP onset increases the difference by 0.5 standard deviation units. However, for the relationship between the Group Difference and T2 time difference relative to T1, whilst not statically significant, the 80% confidence intervals did not overlap with zero, $\beta = 0.0005$ (80% CI $[0.000, 0.001]$), $Q_{(1)} = 1.65$, $p = 0.20$. This indicates that a 1 msec increase in T2 time difference decreases the group difference by 0.0005 standard deviations units; the addition of 500 msec of variability decreases the difference by 0.25 standard deviation units. Considering the exploratory nature of moderator variables in a meta-analysis, we consider this to be suitable for further consideration (*Cumming, 2012*).

## DISCUSSION

In a meta-analysis of attentional blink (AB) experiments focussed on developmental dyslexia, we examined whether the overall between-group difference in T2 accuracy was related to variability in temporal and task-set related features. As noted by *McLean et al. (2010)*, the clearest pattern in this literature is that performance of groups of individuals with dyslexia is poorer overall (see Fig 1D); that is, statistically, there is a main effect, indicative of a general difficulty with the dual-target rapid serial visual presentation (RSVP) paradigm in dyslexia, rather than a specific AB effect. The results of the meta-analysis support this suggestion, with a group difference between −0.70 and −0.74 standard deviation units (see Fig. 2 for a forest plot). Using correlation and moderator linear modelling, pre-RSVP time and T2 temporal variability were identified as explanatory, presentation-related factors for the group difference.

### Pre-RSVP time and endogenous control

The longer the time period before the RSVP onset, the greater the difference between-groups. To be clear, three of the AB and dyslexia experiments included in the meta-analysis were excluded from the analysis of pre-RSVP time because the fixation symbol remained onscreen until a key-press. Therefore, the finding is limited as it is based on eight experiments, with pre-RSVP times of 0, 500, 600, and 1,000 ms. However, in support of the pattern, we have conducted two pilot studies with adults, unselected for reading ability, in which we manipulated the duration of the pre-RSVP time and found that this influences overall performance: shorter intervals corresponding with lower accuracy. Given the strength of the relationship and the results of the pilot studies, pre-RSVP time is worth further consideration.The pre-RSVP time may have a preparatory function for the engagement of *task-set*. Task-set is a cognitive model of the task requirements (*Rogers & Monsell, 1995*; *Monsell, 1996*). This is a similar concept to the visual filter which is implicated in selection theories of the AB (e.g., *Di Lollo et al., 2005*). For a task including two number targets in a series of black letter distractors, the task-set would involve ignoring letters and attending to numbers.

The task-set may also incorporate goals *Ferlazzo et al. (2007)* demonstrated that the AB could be mediated by varying the instructions to participants. With standard instructions, e.g., report the identity of two numbers, an AB effect was observed. However, with modified instructions consisting of a single goal, e.g. report the sum of the numbers, the AB effect was not observed. In relation to temporal effects, the engagement time of task-set may be shorter for a single goal, resulting in more accurate performance.

Temporal expectations of target presentation may be part of task-set with respect to perceptual expectations. When attention can be directed to a particular point in time, decisions regarding a target are more accurate (again see *Nobre, Correa & Coull, 2007*, for a revew). This has also been demonstrated in the AB. When *Martens & Johnson (2005)* cued the temporal location of targets within an RSVP sequence, the AB was reduced. Similar reductions were found when *Badcock et al. (2013)* individually tailored the time before T1 and made it predictable between trials; relative to when the temporal location

of T1 was randomly varied between 250 and 750 ms, the AB was reduced. Further to this, *Tang, Badcock & Visser (2014)* demonstrated that practice with the AB task increased expectations about the temporal locations of the targets. This suggests that, with exposure to RSVP tasks, observers are implicitly generating expectations about the timing of the presentation. This may be a longer learning process in dyslexia (*Badcock, Hogben & Fletcher, 2011*), contributing to lower overall accuracy in the AB.

With respect to the engagement of task-set in dyslexia, the pattern of overall group differences in the AB can be accounted for by a specific difficulty with the endogenous engagement of task-set. Endogenous engagement relates to internally- or observer-driven behaviour, as opposed to exogenous engagement, which relates to stimulus-driven behaviour. The engagement of task-set prior to the RSVP onset requires endogenous control. With limited or no pre-RSVP time, neither group can prepare and the typical readers do not gain the advantage that they do at longer pre-RSVP times.

Through the endogenous engagement of task-set, increasing the pre-RSVP time should enhance preparation for the task and result in better performance (as evidenced by *Badcock et al., 2013*). This was observed in the current data, examining the correlation between overall T2 accuracy and pre-RSVP time (Spearman, single-tailed), but was only reliable in the typical readers: rho = 0.71, $p$ = 0.02; dyslexic readers; rho = 0.22, $p$ = 0.30; consistent with the proposed difficulty with the endogenous engagement of task-set in dyslexic readers. It is important to note that this relationship relates to temporal features of task-set and not to target and distracter categories (*Maki & Padmanabhan, 1994*; *Maki et al., 1997*) or task complexities such as task-switches (see *Potter et al., 1998*), which were not related the group difference for the current data. Experimental manipulation of these factors should be conducted before it can be confidently concluded that categories and complexities have no effect.

## T2 timing and endogenous or exogenous control

Greater temporal variability of T2 relative to T1 was associated with smaller group differences in overall T2 accuracy. This variable was also correlated with the variability of the RSVP duration (.73–.88) as well as the pre-RSVP time (−.77 to −.87). It was calculated as the difference between the maximum and minimum temporal positions of T2 relative to T1. As evident in the preprint of this article (*Badcock & Kidd, 2014*), this variable is highly correlated with the minimum (0.73) and maximum (0.99) T2 temporal positions, so the greater variability is not independent of longer time periods before the presentation of T2. Therefore, this can be considered as T2 timing.

With respect to T2 timing, the engagement of task-set following the RSVP onset can be driven endogenously but will be exogenously influenced by the stimulus, essentially reminding the observer of task requirements. Through the exogenous engagement of task-set, increasing the time before the presentation of a target should enhance preparation for the task and result in better performance (e.g., *Ariga & Yokosawa, 2008*). However, the opposite pattern was observed in the current data: the correlation between overall T2 accuracy and maximum T2 temporal position within the RSVP was negative, Spearman's

rho $= -0.56$, $p = 0.07$. This does not converge with the results for the group difference, suggesting that temporal variability may underpin the relationship between T2 timing and the group difference.

As mentioned, temporal expectations are part of task-set and anticipating when a stimulus will occur improves accuracy (for a review see *Nobre, Correa & Coull, 2007*). With respect to temporal variability: as variability increases, temporal orienting would be poorer, resulting in lower accuracy. This was observed in the current data for both groups, the correlation between overall T2 accuracy and the maximum minus minimum T2 temporal position within the RSVP, (Spearman, single-tailed), but was only reliable in the typical readers: rho $= -0.69$, $p = 0.01$; dyslexic readers; rho $= -0.40$, $p = 0.10$. These results are consistent with a performance advantage for typical readers for presentations with small temporal variability, whereas this does not seem to be the case for dyslexic readers. This advantage may be underpinned or further exaggerated by preparation gained during the pre-RSVP interval.

## Perceptual learning

This interpretation of the findings implicates endogenous control difficulties with dyslexia. To reiterate: this includes poor or slow task preparation and poor or slow tuning of temporal attention to target timings. Perceptual learning, specifically with respect to the Perceptual Anchoring Theory of dyslexia (*Ahissar, 2007*), also offers an account for some aspects of the current findings. A sound perceptual learning system allows observers to take advantage of regularities in psychophysical presentations. By forming 'perceptual anchors' to these regularities, performance improves as the task processing switches from being effortful to automatic. For the current results, repetition of the pre-RSVP time or small temporal variability in T2 position may have been utilised by typical observers to improve accuracy, whereas this is not the case in dyslexia (but see *Ziegler, 2008*). An anchoring deficit in dyslexia would account for limited performance improvement with increasing pre-RSVP times, as dyslexic readers would not form an anchor to this regularity. Similarly, dyslexic readers may not take advantage of small temporal variability in T2 position, whereas it may be difficult for any observer, sound anchoring or otherwise, to anchor to larger temporal variability, as was observed in the typical readers. We do note that perceptual anchoring does not account for the increasing group difference with increasing pre-RSVP time (e.g., a preparatory component); however, the methods and theories relevant to perceptual anchoring may prove useful in investigating the current suggestions.

A major caveat to the above suggestions is that meta-analytic techniques stepping beyond tests of homogeneity are exploratory in nature. The variables have been highlighted within inherently complex paradigms conducted in multiple laboratories, each with subtly different goals in mind. Therefore we raise these variables, and subsequent discussion, as speculation that requires empirical interrogation.

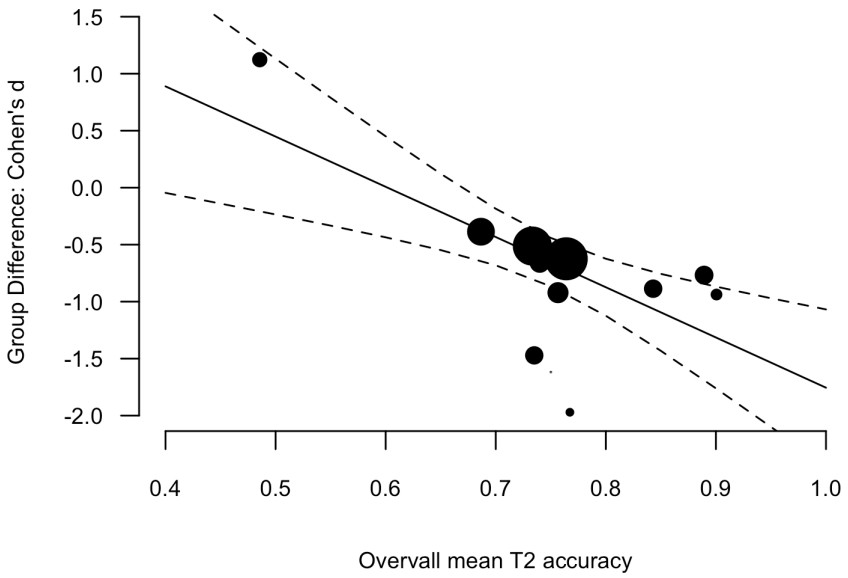

**Figure 4** **Scatter plot and linear regression fit between the Group Difference (Cohen's d, *y*-axis) and Overall mean T2 accuracy (mean of the two groups).** Data point size is based upon the inverse of the variance (i.e. larger points correspond to smaller variance), solid lines are regression fits, and dashed lines are 95% confidence intervals.

## FURTHER CONSIDERATIONS

### The Lacroix et al. anomalous result

There is one anomalous result in the AB and dyslexia literature. *Lacroix et al. (2005)* reported better overall performance in a group of children (15-years of age) with developmental dyslexia, relatively to typically reading peers. The task involved the identification of red number targets in black number distracters. *Buchholz & Aimola Davies (2007)* conducted the same design in adults and found the typical pattern of results. *Lallier et al. (2010)* conducted a T1 identification-T2 detection variation in younger children (approximately 10-years of age) and also found the typical pattern of results. One feature of the Lacroix et al. results that stands out is that the overall T2 accuracy (i.e., the mean of the two groups) was low (49% with next lowest, 73%). When considered as a predictor of the Group Difference including the Lacroix et al. result, overall T2 accuracy was a significant moderator; $\beta = -4.41, [-7.01, -1.80], Q_{(1)} = 10.99, p < 0.001$ (see Fig. 4).

It is possible that at this level of difficulty, observers may adopt a different strategy, which may not necessarily be conscious. *Lacroix et al. (2005)* suggest that superior dual-target accuracy in the dyslexic group may be explained by development differences in the depth of target processing. Suppose the group with dyslexia have less detailed representations for numbers and the group with typical reading have more detailed representations for numbers. If greater resources were required to access the more detailed representations, the typical readers would exhibit poorer performance. In order to best integrate this result with the other literature, replication reducing overall accuracy may be an important step.

## Confounded variables

There were significant inter-correlations between the variables, which are due to the commonalities and differences between experiments. For example, pre-RSVP time and total identities were correlated with T2 temporal variability. These are impossible to disentangle in the current dataset. Without direct experimental manipulation of these variables, it cannot be determined whether the highlighted variables are critical factors. Therefore we have purposefully restricted the considerations of the relationship between the highlighted variables and reading mechanisms until such a relationship is demonstrated.

One suggestion arising from these confounds is that a common paradigm be adopted for AB and dyslexia research. Considering the limited reliance on literacy experience for shape-targets and random-dot distracters used by *McLean et al. (2010)* and *Visser, Boden & Giaschi (2004)*, this seems like a good starting place. Furthermore, this paradigm has also been used to evaluate the relationship between AB performance and typical reading (*McLean et al., 2009*; *La Rocque & Visser, 2009*), so there is a growing body of evidence common to this target and distracter set related to reading.

## Subtypes of dyslexia

We have not been able to address subtypes of dyslexia (*Jones, Castles & Kohnen, 2011*; *Coltheart & Kohnen, 2012*) in this meta-analysis and we want to flag this as an important consideration for future research. *McLean et al. (2010)* examined their results with respect to non-word and irregular word reading independently but did not find a difference with respect to the between group relationship. *La Rocque & Visser (2009)* found a relationship between non-wording reading and AB magnitude in typically reading adults, suggesting some role of phonological processing. However, this specifically related to a between group interaction rather than an overall main effect, therefore it is not clear what role phonological processing would have for the current set of variables. The main effect is also evident in shallow orthographies (i.e., Italian, *Facoetti et al., 2008*) in which the letter-sound correspondences tend not to underpin reading difficulties; therefore, exploration of this relationship should not be limited to phonological deficits, and attentional subtypes (*Friedmann, Kerbel & Shvimer, 2010*; *Kohnen et al., 2012*) and capacities (*Valdois et al., 2003*; *Bosse, Tainturier & Valdois, 2007*) should also be examined. A critical step in pinning down the relevance of any variable in relationship to dyslexia is pinning down which particular component of the reading process, and in turn which subtype, the variable is associated with.

## The available evidence

This research is limited by the available evidence. The number of experiments published in the literature is small, and exclusory criteria reduced this to 11 (8 for one variable) in the current meta-analysis. The dataset is limited to published work which may be associated with biases with respect to the publication of significant effects. The results should be interpreted with these limitations in mind.

## SUMMARY AND CONCLUSION

In this paper we report on a meta-analysis of published AB experiments conducted in developmental dyslexia. We examined the common occurrence of an overall between-group difference where dyslexic readers exhibit lower target reporting accuracy in relation to task parameters which varied between and within experiments. This between-group difference was related to pre-RSVP time and second target temporal variablity. Future investigations should consider the endogenous engagement of task-set and temporal learning in the AB and dyslexia in order to best determine the relationship between reading and the AB, and whether it may be a potential tool for intervention.

## ACKNOWLEDGEMENTS

The authors would like to thank Veronika Coltheart for her comments on an earlier version of the manuscript, Gen McArthur for her valuable input and support, as well as Sachiko Kinoshita and Maree Tyson-Parry for their ears and thought-provoking discussions. We would also like to thank all the authors of the AB and dyslexia experiment for sharing their data and details of their research procedures.

### Funding

This research was funded by the Australian Research Council Centre of Excellence in Cognition and its Disorders (CE110001021). The funders had no role in study design, data collection and analysis, decision to publish, or preparation of the manuscript.

### Grant Disclosures

The following grant information was disclosed by the authors:
Australian Research Council: CE110001021.

### Competing Interests

The authors declare there are no competing interests.

### Author Contributions

- Nicholas A. Badcock and Joanna C. Kidd conceived and designed the experiments, analyzed the data, contributed reagents/materials/analysis tools, wrote the paper, prepared figures and/or tables, reviewed drafts of the paper.

### Supplemental Information

Supplemental information for this article can be found online at http://dx.doi.org/10.7717/peerj.746#supplemental-information.

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
