# Peer review of "Temporal variability predicts the magnitude of between-group attentional blink differences in developmental dyslexia: a meta-analysis"

_PeerJ, doi:10.7717/peerj.746_

## Round 0.1 · original submission · Major Revisions

Dear Authors,

Could you do the necessary revisions as per the comments by the two reviewers especially those from Reviewer 1, who akes several suggestions to improve the quality of the manuscript.

·

Basic reporting

• The acronym RSVP is often used in ways that seem to mean something other than « rapid serial visual presentation ». Sometimes to refer to the fixation symbol at the beginning of a trial, sometimes to refer to a trial or to items within a trial. The language should be clarified in this respect. My impression is that removing RSVP in many instances will help.
• Line 153 the fixation duration parameter should be better explained. It is actually the fixation symbol duration. It remains unclear to me what a duration of 0 ms means. If it means “no fixation symbol”, is it legitimate to treat this situation on the same scale as those with a fixation symbol?
• Line 161: “if T2 is a FIXED letter of the alphabet” would be clearer.
• Line 163: there -> they
• Line 187: remove “in”
• Lines 231-232: r values differ slightly from those reported in the table.

Experimental design

Although I am not an expert of meta-analyses (I hope that another reviewer is), I have a few important concerns with the way the meta-analysis is conducted in the present paper :
• Three studies are excluded from the meta-analysis, two of which for reasons that seem hardly justified (lines 128-135). One is excluded on the grounds that the effect is in the opposite direction from the others. In my view this is no reason for exclusion. Imagine if a meta-analysis of clinical trials excluded trials where the treatment had a negative (rather than positive) effect on patients! The virtue of meta-analyses is to provide an objective synthesis of contradictory research findings. It defies the entire purpose of a meta-analysis to a priori exclude a study because it seems in contradiction with the others. The same reasoning applies for the second study that is excluded on the grounds that the authors judge the effect size to be anomalous. If they suspect that the odd effect size results from a typo in a table or a miscalculation, then they should contact the authors to clarify this. But if it turns out that the numbers do reflect the actual data, then the study should be included in the meta-analysis. With respect to the third study (whose effect size could not be computed from the reported results), it should be easy enough to contact the authors and obtain the desired numbers. Given that the number of studies in this meta-analysis is quite low for the purpose of the mediation analyses proposed, all reasonable efforts should be made to include as many as possible.
• I wonder if the approach consisting of just testing simple correlations between effect size and all possible parameters represents the state of the art for this kind of research. First, shouldn’t a test of heterogeneity (of effect sizes) be done to justify the search for mediators? Second, shouldn’t slightly more sophisticated meta-regression models be used? Again, I am not an expert on meta-analysis so I might be wrong on this. The view of a real expert would be welcome.
• The number of parameters investigated (18) is disproportionately large compared to the number of studies meta-analysed (6 to 9). Given that many parameters are confounded with each other, maybe a more parsimonious approach would be to start with a factor analysis, use it (if its results are clear enough) to compute a more limited number of composite measures (average z-scores of some variables) to be tested as mediators in the meta-regression, and when some of them turn out to be significant, discuss which of the underlying variables is likely to produce the effect. It may well be that in some cases this is undecidable because some parameters are equally plausible and turn out to be confounded in published studies. This approach may be preferable to adjudicating between parameters on the basis of small, non-significant differences between correlations (as has been done between distractor ID and T2 time max and difference, for instance). An alternative approach would be to carry out multiple rather than simple regressions, in order to try and disentangle between confounded variables. But this will still be limited by the small number of studies.
More minor comments:
• The justification for rescuing the non-significant effect of SOA with R=-0.66 but not the non-significant effect of distractor ID with R=-0.78 is a bit awkward. Either Pearson’s test applies or it doesn’t. Perhaps reporting and reasoning only on Pearson when distribution is normal, and on Spearman when not, would make it clearer. This problem would disappear with the factor analysis approach mentioned above.

Validity of the findings

Apart from the problem of confounded variables that are impossible to disentangle, I am not entirely convinced by some of the interpretations offered in the discussion.
With respect to fixation duration, wouldn’t the authors’ interpretation predict a negative correlation instead? If the task is more difficult with shorter fixation duration, and if dyslexics need more preparation time, they should be at a greater disadvantage (large effect size) when fixation duration is short (thus negative correlation). The idea that dyslexics would not benefit at all from greater preparation does not seem very plausible.
Similarly for T2 temporal position and variability, it is suggested that in the more variable and difficult conditions, the task is so challenging that floor effects are reached for both groups (hence small effect size), and it is in the less variable (easier) conditions that group differences emerge. In many tasks with dyslexic participants, this is just the contrary: performance is at ceiling in easy conditions and group differences emerge in more difficult conditions (see for instance discussion of task difficulty in Ramus and Ahissar 2012). Furthermore the Badcock et al (2011) study that is cited does not seem to support the authors’ interpretation: if there is a group difference in the first half of the experiment (without practice, difficult) but not in the second half (with practice, easier), then this is a ceiling, not a floor effect. At any rate, any conjecture on a floor or on a ceiling effect could be tested by looking at absolute performance in the initial studies.
I can’t see clearly the link between SOA and the discussion in lines 349-365.
In their discussion (section 7.1) of the “anomalous” results of Lacroix et al. and Buchholz et al., the authors speculate about experimental parameters that might explain such results: children rather than adult participants, number stimuli, target-distractor relationship, and task-set. But the proper way to test these hypotheses would be to include these studies and these additional parameters in the analysis.
Line 390: I don’t understand how the effect might reverse (as opposed to just disappearing) in adults as retrieval becomes more automatic.

Reviewer 2 ·

Basic reporting

No Comments

Experimental design

No Comments

Validity of the findings

No Comments

Additional comments

minor clarification: Method in abstract states 13 experiments (11 papers) returned when search conducted, but experimental section states 26 entries returned in search. Otherwise a very well written and informative manuscript

---

## Round 0.2 · Minor Revisions

Dear Authors,You manuscript will need to undergo minor revisions.Please do the revisions as soon as possible as the revised manuscript will need to undergo another round of peer review.

·

Basic reporting

Abstract: I don't think it is the test of homogeneity of variances that yields the group difference.
line 40: I would remove "as it ostensibly does in dyslexia", as this is a controversial statement that would require justification, and here it's not crucial.
line 43: regressive eye movements may occur in dyslexic individuals simply because they read less well, so this may have nothing to do with attentional blink.
Method: it is odd to present all the moderators before the section on group difference (contrary to the results section).
line 129: these experiments were excluded FROM THIS ANALYSIS; It would be good to explicitly state on how many experiments this moderator is analysed.
line 183: it would be good to add to this section a sentence about how moderation was tested.
line 224: it would be helpful to spell out the interpretation of beta values. Is it that each extra millisecond of pre-RSVP time increases the group difference by 0.001 d? So one second pre-RSVP time for one standard deviation of group difference? Same for the other moderator.
line 227: the interval does not include the beta value.
line 247: fixation should probably be replaced with time.
line 272: accuracy -> accurate
line 303: why are the rho values negative? This seems contrary to the explanation in the text.
346 and following: a discussion of subtypes is welcome but should not be limited to phonological vs. surface subtypes of reading disability, whose validity is debatable. It would seem more relevant to discuss phonological (in the broad sense of a phonological deficit, not of a deficit in the phonological route of the reading system) vs. visual/attentional subtypes.
Table 1: add a column with the number of experiments contributing to each moderator.
Supplementary material: I think that the supplementary table contains useful enough information to be integrated in the article. In fact it could be merged with Table 1, with mean, median and Shapiro-Wilk statistic (and nb of experiments as indicated above) listed as additional lines to the supplementary table.

Experimental design

OK

Validity of the findings

Overall I find that it remains difficult to wrap one's head around the results: why are these moderators significant and why do they seem to act in opposite directions? One reason may be that the real moderators are variables correlated with the current ones. On line 257, the authors suggest that T2 temporal variability might be a proxy for T2 timing. If they think that T2 timing is more interpretable, then why not use that variable as a moderator in the first place? After all, why analyse moderators for which there would be no plausible interpretation, when they are highly correlated variables that would be more easily interpreted?
In a similar vein, the authors go on to interpret their moderators in terms of task set. This is quite troubling given that task set complexity was one of their moderators and that it was not significant. Of course, in a sense, all the moderators reflect some aspect of task set. But then one still needs to explain why this particular timing aspect of task set affects the group difference more than others.
Now it occurs to me that the puzzling direction of pre-RSVP time is compatible with Ahissar's anchoring deficit theory. Here basically dyslexics fail to benefit from task-specific regularities that benefit controls. May be worth discussing. But this cannot explain the opposite effect of T2 temporal variability.

Additional comments

This new version of the paper is much improved and largely satisfactory. I mostly have minor comments. The discussion is the main area that could still be improved.

---

## Round 0.3 · accepted · Accept

Dear Authors,

This manuscript after two revisions is now suitable for publication in Peer J.Thank you for your patience in attending to the comments of the peer reviewers so as this manuscript could be made more citable as well as the experiment methodology more reproducible in other labs around the world.